# Point-of-Care Lung Ultrasound in the Intensive Care Unit—The Dark Side of Radiology: Where Do We Stand?

**DOI:** 10.3390/jpm13111541

**Published:** 2023-10-26

**Authors:** Marco Di Serafino, Giuseppina Dell’Aversano Orabona, Martina Caruso, Costanza Camillo, Daniela Viscardi, Francesca Iacobellis, Roberto Ronza, Vittorio Sabatino, Luigi Barbuto, Gaspare Oliva, Luigia Romano

**Affiliations:** 1Department of General and Emergency Radiology, “Antonio Cardarelli” Hospital, 80131 Naples, Italy; giuseppina.dellaversanoorabona@aocardarelli.it (G.D.O.); martina.caruso@aocardarelli.it (M.C.); costanza.camillo@aocardarelli.it (C.C.); francesca.iacobellis@aocardarelli.it (F.I.); roberto.ronza@aocardarelli.it (R.R.); vittorio.sabatino@aocardarelli.it (V.S.); luigi.barbuto@aocardarelli.it (L.B.); gaspare.oliva@aocardarelli.it (G.O.); luigia.romano@aocardarelli.it (L.R.); 2Department of Intensive Care and Resuscitation, “Antonio Cardarelli” Hospital, 80131 Naples, Italy; daniela.viscardi@aocardarelli.it

**Keywords:** intensive care unit, chest X-ray, lung ultrasound, point-of-care ultrasound

## Abstract

Patients in intensive care units (ICUs) are critically ill and require constant monitoring of clinical conditions. Due to the severity of the underlying disease and the need to monitor devices, imaging plays a crucial role in critically ill patients’ care. Given the clinical complexity of these patients, who typically need respiratory assistance as well as continuous monitoring of vital functions and equipment, computed tomography (CT) can be regarded as the diagnostic gold standard, although it is not a bedside diagnostic technique. Despite its limitations, portable chest X-ray (CXR) is still today an essential diagnostic tool used in the ICU. Being a widely accessible imaging technique, which can be performed at the patient’s bedside and at a low healthcare cost, it provides additional diagnostic support to the patient’s clinical management. In recent years, the use of point-of-care lung ultrasound (LUS) in ICUs for procedure guidance, diagnosis, and screening has proliferated, and it is usually performed at the patient’s bedside. This review illustrates the role of point-of-care LUS in ICUs from a purely radiological point of view as an advanced method in ICU CXR reports to improve the interpretation and monitoring of lung CXR findings.

## 1. Introduction

The anterior–posterior supine chest X-ray (CXR) is a routine investigation performed in intensive care units (ICUs). Being a widely accessible imaging technique, which can be performed at the patient’s bedside and at a low healthcare cost, it provides additional diagnostic support to the clinical examination of patients who usually require respiratory assistance and constant monitoring of vital functions and devices [1]. However, its limitations in terms of technical and diagnostic performance are equally well known when compared to both a standard CXR and the better performing and more panoramic chest computed tomography (CT) scan, with image quality often resulting in a frustrating experience for both the clinician, who has to manage a critical patient, and the radiologist, who has to provide the best diagnostics guidance [2]. The technical limitations of the image produced are mitigated by the widespread availability of sufficiently powered X-ray tubes, image digitization, and a rigorous execution technique. Nevertheless, interpretative shortcomings in the cardiopulmonary findings obtained remain, which are unavoidable and inherent in the method [1,2,3,4]. If on the one hand the chest CT scan can be considered the diagnostic gold standard in view of the critical state of these patients, it is quite understandable that, since they are critical patients, they cannot be easily transported to the diagnostic radiology departments in order to undergo a chest CT scan [5,6,7].

In addition, the frequency of diagnostic procedures required for these patients, often with daily monitoring of the chest, is to date still a widely controversial issue, and anterior–posterior supine CXR performed at the patient’s bedside is still the best solution [8,9,10].

The well-established role of bedside lung ultrasound (LUS) performed by the resuscitation team itself represents a crucial turning point in the diagnosis and management of the critically ill patient as well as a guide to the interventional procedures to be performed at the patient’s bedside, such as pleural drainage or the insertion and monitoring of devices (Figure 1) [11,12,13].

LUS is thus widely accepted in several codified clinical management protocols, as summarized in Table 1 [5,14,15].

Furthermore, the specificity and sensitivity of LUS when compared with anterior–posterior supine CXR are encouraging enough to make it an attractive and often alternative choice to CXR (Table 2) [4,5,16].

All this is also amplified in view of the method’s proximity to resuscitation areas, its high clinical value, and the absence of the use of ionizing radiation, albeit settling for a partial view, limited to artifactual deductions of the subpleural lung parenchyma, which is highly subjective with the risks of operator dependence [4]. The integration of these methods, i.e., CXR and LUS, could be an incisive turning point, especially in complex clinical settings such as ICUs, where management decisions must be made rapidly and coordinated among the various specialists [17,18].

In this imaging review, we provide some food for thought by drawing on our organizational expertise in interpreting CXR findings and lowering the number of serial CXR checks through the use of LUS as a clinical radiology point of care. This is conducted solely to supplement the CXR findings, monitor those that are readily accessible to the investigation itself, and reduce the produced radiant dose. Finally, but just as importantly, we focus more on the CXR reading through a more insightful echo-mediated clinical interpretation (Figure 2).

Table 3 shows the main pleural–parenchymal complications observed in ICUs that are the subject of this imaging review [19].

They are divided into three specific paragraphs where the use of LUS, as a methodological advance in the bedside CXR reports in ICUs, is crucial for diagnosing and monitoring areas of decreased transparency around the chest found on X-rays (atelectasis, pneumonia, pleural effusion, pulmonary edema, acute respiratory distress syndrome (ARDS), and pulmonary contusions), in pneumothorax and device monitoring.

## 2. Characterizing and Monitoring Areas of Decreased Transparency on the CXR

LUS allows us to read the CXR by identifying and better characterizing areas of decreased transparency found on X-rays and also defining their solid or liquid nature and assisting drainage procedures (Figure 3 and Figure 4) [28,29,30,31,32].

However, the initial CXR is still an essential diagnostic tool to obtain an overview of pleural–parenchymal—but not necessarily subpleural—findings and cardio-mediastinal findings, just like an “open window to the chest”. It is then supplemented by a targeted LUS investigation where these findings are available to the investigation itself, in order to make the diagnosis and undertake a correct evolutionary/resolutive monitoring in line with the clinical course of the patient being treated [17,18]. Furthermore, the potential role of contrast-enhanced ultrasound (CEUS) cannot be excluded [33,34,35]. CEUS could be a valuable diagnostic tool in differentiating peripheral areas of parenchymal-enhancing consolidation, such as pneumonic foci, atelectasis, or tumors from peripheral areas of non-enhancing parenchymal infarction [33,34,35]. However, dynamic CEUS parameters cannot effectively differentiate between benign and malignant peripheral pulmonary lesions due to an overlap of CEUS timings and patterns [36]. On the other hand, CEUS plays an additional role in diagnosing pleuritis or empyema and in guiding their drainage or the biopsy of solid lesions [33,34,35]. Figure 5 illustrates an example of CEUS diagnostic utility for LUS.

### 2.1. Atelectasis

The term atelectasis refers to the collapse of groups of alveoli resulting in a concurrent decrease in intrapulmonary airflow sustained by a dysfunction of the diaphragm, causing respiratory impairment, as in post-operative adhesions [37]. The main physiologic causes of atelectasis are compression of lung tissue, absorption of alveolar air, and impairment of surfactant function [37,38,39].

Compression atelectasis occurs when the transmural pressure is reduced to a level that allows the alveolus to collapse. The supine position, positive-pressure ventilation, and muscle paralysis, when applied in ICUs, cause a cephalad shift of the diaphragm, which normally permits differential pressures in the abdomen and chest, resulting in a concurrent decrease in intrapulmonary airflow. In an anesthetized patient a cephalad diaphragm displacement, differential regional diaphragmatic changes, a shift of thoracic central vascular blood into the abdomen, and an increase in regional pleural pressure result in negative transpulmonary pressure and compressive atelectasis [37,38,39].

Resorption atelectasis or gas atelectasis stems from the resorption of gas from alveoli when communication between the alveoli and the trachea is obstructed. The obstruction leads to a non-ventilation of the distal airways; the gas residing in that region is completely absorbed by pulmonary blood flowing through that area. Aspirated foreign bodies, food and gastric contents, malpositioned endotracheal tubes, and mucous plugs favor the appearance of resorption atelectasis in ICUs [37,38,39]. Resorption atelectasis can also be seen in acute bronchitis and pneumonia from the obstruction of small bronchi and bronchioles by inflammatory exudate and tumors (bronchogenic carcinoma, bronchial carcinoid, metastastates, lymphoma) [37,38,39].

A number of conditions impair muco-ciliary clearance: thoracic and abdominal pain, thoracic and abdominal surgery or trauma, central venous system depression, respiratory depressant medication, anticholinergic medication, general anesthesia, endotracheal intubation, ventilation with dry gases, inspiring oxygen in higher concentrations. Impairment of muco-ciliary transport causes the pooling of retained secretions in the smaller airways, and it favors the appearance of atelectasis. In the post-operative patient and in patients recovering in the ICU many factors are combined [40].

Development of atelectasis is associated with the development of several pathophysiologic effects, including decreased compliance, impairment of oxygenation, increased pulmonary vascular resistance, and development of lung injury. Atelectasis produces alveolar hypoxia and pulmonary vasoconstriction to prevent ventilation–perfusion mismatching and to minimize arterial hypoxia [37,38,39,40]. Atelectasis itself is often asymptomatic unless hypoxemia or pneumonia develops. Symptoms of hypoxemia tend to be related to acuity and extent of atelectasis. Dyspnea or even respiratory failure can develop with rapid, extensive atelectasis [37,38,39,40].

The main CXR sign is a lobar or sub-lobar parenchymal thickening often evident at the lower lobes with partial or total disappearance of the profile of the hemidiaphragms or cardiac silhouette and also with the mediastinum that is shifted toward the collapsed lung area [41,42].

On LUS, the common finding is the sign of tissue, the so-called pulmonary “hepatization”, suggesting collapsed parenchyma with air bronchograms frequently static or unaffected by respiratory dynamics as they are trapped in the area of consolidation; the bronchi are often in a parallel arrangement, unlike the typical tree-like arrangement of the pneumonia consolidation, in line with a reduction in lung volume [5,21,29,30,38,43]. Furthermore, in the collapsed parenchymal area the “lung sliding” is usually absent while the “lung pulse” appears more represented, which is a sign of the perception of the impact of cardiac activity on the collapsed lung [44].

Figure 6 is an example of CXR and LUS diagnostic integration in detecting atelectasis.

### 2.2. Pneumonia

Pneumonia ranks among the more common nosocomial infections [45,46,47]. The mechanisms of contamination may depend on the aspiration of active flora in the oropharynx, inhalation of aerosols containing pathogenic microorganisms, or more rarely, hematogenous spread to the lungs [45,46,47].

In ICUs, hospital-acquired pneumonia (HAP) and ventilator-associated pneumonia (VAP) occur more frequently. HAP is a new pneumonia (a lower respiratory tract infection verified by the presence of a new pulmonary infiltrate on imaging) that develops more than 48 h after admission in non-intubated patients. VAP, the most common and fatal nosocomial infection in critical care, is a new pneumonia that develops after 48 h of endotracheal intubation. Importantly, by the time of VAP onset, patients may have already been extubated [45,46,47]. Aspiration is an important contributor to the pathogenesis of HAP and VAP. Further, proton-pump inhibitors and histamine-2 receptor blockers, by suppressing acid production, can allow nosocomial pathogens to colonize the oropharynx and endotracheal tube and be aspirated [45,46,47]. Owing to their low sensitivity and specificity, history and physical examination are considered suboptimal to confirm or exclude the diagnosis.

Guidelines recommend that a new pulmonary infiltrate at the CXR should be present to classify a hospitalized patient as having the diagnosis of pneumonia. Oxygen saturation (SpO_2_) and arterial gas analysis can provide important information about severity. In an attempt to determine whether a typical pathogen is the etiology of pneumonia, all patients should have a sputum specimen for Gram stain and culture as well as two sets of blood cultures obtained before the institution of antimicrobial therapy. Fluid samples from the pleural space and bronchoscopy represent invasive tests that are also useful in pneumonia diagnosis [45,46,47].

The CXR shows the presence and extent of the condition, any concurrent cavities, and the appearance of parapneumonic effusions or abscesses [41,42].

On LUS examination, areas of subpleural consolidation with a hypoechogenic echostructure, irregular margins, and of various shapes and sizes with a tendency to confluence can be found, often associated with the detection of dynamic air bronchograms or hyperechogenic punctiform foci representing air in the bronchi that moves in line with the respiratory excursion. Furthermore, if a significant peripheral abscess cavitation develops, it appears on the LUS as a colliquated central lacuna with debris and gas or with a pseudo-solid appearance related to suppuration development [5,21,29,30,43].

Figure 4, Figure 7, and Figure 8 show some examples of CXR and LUS diagnostic integration in detecting pneumonia.

### 2.3. Pleural Effusion

A pleural effusion is defined as an excess of pleural fluid within the pleural space. It represents a disturbance of equilibrium between production and resorption. Normally, a small amount of pleural liquid (5–10 mL, 0.1 mL/kg per hemithorax in healthy adults) exists within the pleural space [48]. The volume of the pleural fluid is determined by the balance of the hydrostatic and oncotic pressure differences that are present between the systemic and pulmonary circulation and the pleural space. Both the visceral and the parietal pleura play an important role in fluid homeostasis in the pleural space; especially, the parietal side of the pleura accounts for most of the production of pleural fluid and for most of its resorption [48,49,50]. Excess fluid accumulation in the pleural space can be caused by both benign and life-threatening conditions. A transudate occurs when systemic factors influencing the formation and absorption of pleural fluid (hydrostatic and oncotic pressures) are altered so that fluid accumulates. The reason to make this differentiation is that the existence of a transudative pleural effusion indicates that systemic factors such as heart failure or cirrhosis are responsible for the effusion, with the existence of examples such as congestive heart failure, cirrhosis, nephrotic syndrome, urinothorax, hypoalbuminemia, and cerebrospinal fluid leak [48,49,50].

An exudative effusion indicates that local factors are responsible for the effusion. Malignant disease (carcinoma of any origin, but especially lung and breast; lymphoma; mesothelioma); infections (parapneumonic effusion; tuberculous pleurisy; fungal, parasitic or viral infections); autoimmune inflammatory diseases (systemic lupus erythematosus and other connective tissue diseases; rheumatoid arthritis); pulmonary embolism; intra-abdominal processes (pancreatitis; subphrenic/hepatic abscess); drugs (amiodarone, methotrexate, nitrofurantoin and others); traumatic hemothorax; cardiac bypass surgery; post-cardiac injury syndrome; and post-radiation therapy could cause exudative effusion [48,49,50].

Clinical assessment to determine the appropriate investigation and management is essential. Although symptoms specific to the underlying cause may be present, pleural effusions usually present with nonspecific symptoms such as dyspnea, cough, and pleuritic pain. Such symptoms, if present, reflect an inflammatory response of the pleura, a restriction of pulmonary mechanics, or a disturbance of gas exchange; the severity of these symptoms depends on effusion size and the patient’s cardiopulmonary reserve [48,49,50]. The most common symptom of pleural effusion is dyspnea. The severity of dyspnea is only loosely correlated with the size of the effusion, and it could require patient endotracheal intubation and hospitalization in the ICU.

Detection of a pleural effusion by examination is determined by its size. A pleural effusion of less than 300 mL is likely to be clinically undetectable, whereas a large effusion (>1500 mL) may cause significant hemithorax asymmetry. Chest examination typically reveals dullness to percussion, the absence of fremitus, and diminished breath sounds or their absence. Distended neck veins, an S3 gallop, or peripheral edema suggests congestive heart failure, and a right ventricular heave or thrombophlebitis suggests pulmonary embolus [48,49,50].

Moderate effusion often goes undetected by CXR, especially by supine CXR, which can document abnormalities when the amount of pleural effusion reaches 175–525 mL; therefore, only when the amount of fluid increases, the hemithorax becomes hazy, the pulmonary vessels become less recognizable, and the profile of the hemidiaphragm fades and becomes unrecognizable [28,41,42].

According to this evidence, the LUS plays an important role in the identification and quantification of small pleural effusions not detectable by CXR. Simple transudate fluid collections are evident on LUS as anechogenic spaces between the pleural layers; debris that moves on respiratory movement may also be present. A more compact septated collection with a tendency to sacculate may indicate a pleural empyema [5,26,29,30].

LUS may also be a guiding imaging tool both for diagnostic pleural drainage, rather than for therapeutic pleural drainage or thoracentesis, and for follow-up procedures such as iatrogenic pneumothorax occurrences [26,50].

Figure 9 and Figure 10 show examples of CXR and LUS diagnostic integration in detecting pleural effusion.

### 2.4. Cardiogenic Pulmonary Edema

Pulmonary edema is defined as increased fluid in the interstitial and/or alveolar spaces of the lung parenchyma. Acute cardiogenic and/or volume overload edema usually have increased pulmonary venous pressure because of elevations in left atrial and left ventricle end-diastolic pressure [51,52].

The most frequent cause of congestive heart failure is myocardial infarction, although heart failure has many causes, including hypertension, myocarditis, acute onset cardiomyopathy, left ventricle dysfunction or failure, papillary muscle dysfunction, ventricular septal rupture, acute severe aortic insufficiency, acute severe mitral regurgitation, arrhythmias, myocardial contusion/post-cardiac arrest stunning, cardiac tamponade, toxic and metabolic, beta block or calcium channel antagonist overdose, and pheochromocytoma [51,52].

Clinical presentation is characterized by shortness of breath at rest and worsens with exertion, tachypnea, tachycardia, and relative hypoxemia. The lung exam reveals crackles or gasps. Jugular venous pressure may be normal or increased. The S3 gallop on the electrocardiographic exam is relatively specific for cardiogenic edema. A focused exam for cardiac murmurs consistent with valvular stenosis, regurgitation, or signs of right heart failure is of high importance [51,52].

On the CXR, when the edema is still confined to the interstitium, it occurs as a thickening of the associated spaces, especially in the declivities of the lungs where the hydrostatic pressure is higher, so that a thickening of the interlobular septa or so-called Kerley lines, thickening of the walls of the bronchi, shading of the vessel profile, and thickening of the subpleural interstitium can be detected [41,42]. As the edema worsens, vascular and bilateral cotton wool opacities with no aero-bronchogram appear [41,42]. Usually, the heart is enlarged and pleural effusion coexists [41,42].

Multiple B-lines tending to confluence (so-called white lung) are evident on LUS depending on the severity of the condition and are associated with pleural effusion, inferior vena cava ectasia, and heart failure [5,14,15,29,30].

Figure 11 shows an example of CXR and LUS diagnostic integration in detecting cardiogenic pulmonary edema.

### 2.5. Acute Respiratory Distress Syndrome

ARDS is a syndrome of acute respiratory failure caused by non-cardiogenic pulmonary edema representing the most severe stage of acute lung damage due to alveolar–capillary endothelial damage with leakage of fluid from the intravascular to the extravascular compartment [53]. ARDS accounts for 10% of ICU admissions, representing more than 3 million patients with ARDS annually [53,54]. ARDS progresses through several phases (exudative, proliferative, and fibrotic) after a direct pulmonary or indirect extrapulmonary insult, and it is classified as low, moderate, and severe according to Berlin criteria [53,54].

ARDS initially manifests as dyspnea, tachypnea, and hypoxemia, then quickly evolves into respiratory failure. Identification of a specific cause for ARDS remains a crucial therapeutic goal to improve outcomes associated with ARDS. There is still no effective pharmacotherapy for this syndrome, and the treatment remains primarily supportive and includes mechanical ventilation, fluid management strategy, prophylaxis for stress ulcers and venous thromboembolism, nutritional support, and treatment of the underlying injury [53,54].

The CXR shows the presence of more or less homogeneously distributed patches at an already advanced stage in both lungs with less gravitational tendency and less confluence as well as evidence of an internal air bronchogram [41,42]. It may regress, evolve into fibrosis, or in adverse cases show a mosaic evolution with temporal changes in opacities [41,42].

On a LUS, multiple B-lines with irregular distribution alternate with areas of hypoechogenic subpleural consolidation with concurrent air bronchograms [29,30]. The pleural line is also frequently irregular [29,30,31]. Furthermore, the possibility of highlighting healthy lung areas allows for orientation in the differential diagnosis of hydrostatic pulmonary edema [29,30,31].

Figure 12 shows an example of CXR and LUS diagnostic integration in detecting and monitoring ARDS.

### 2.6. Pulmonary Contusion

A pulmonary contusion is an entity defined as an alveolar hemorrhage caused by an injury to the alveolar capillaries and pulmonary parenchymal destruction after blunt chest trauma. It is a common finding after mechanisms of injury that impart substantial kinetic energy to the thorax, and often the rate of pulmonary contusion is linearly associated with the severity of injury to the bony thorax. Lung tissue injury occurs when the chest wall bends inward following the trauma [22,55]. The possible mechanisms are based on inertial effects, mainly on different tissue densities, light alveolar tissue, and heavy hilar structures. Rib fractures and their degree of displacement, as well as flail chest and penetrating mechanisms, contribute to the severity of the underlying lung injury [22,55].

Parenchymal lung injury leads to pathophysiologic changes, the severity of which depends on the extent of injury. The number of affected lobes has been reported to determine the outcome. The physiologic consequences of alveolar hemorrhage and pulmonary parenchymal destruction typically manifest themselves within 24–48 h of injury and usually resolve within approximately 7–14 days, but delayed deterioration may occur [22,55].

Clinical manifestations include increased work at breathing, respiratory distress with hypoxemia and hypercarbia. Patients may present with a rapid respiratory rate, rhonchi or wheezes, or even hemoptysis. Bleeding into uninvolved lung segments may cause bronchospasm and may compromise alveolar function [22,55].

On the CXR, pulmonary contusions appear as diffuse or patchy parenchymal opacities. However, CXRs have a low sensitivity in detecting foci of contusion, in view of the possible coexistence of hemothorax or pneumothorax [41,42].

In view of the fact that the areas of contusion are predominantly subpleural in extension and at the site of impact, LUS may document and monitor obvious foci of contusion as hypoechogenic areas with poorly defined margins often coexisting with multiple confluent B-lines [22,29,30].

Figure 13 shows an example of CXR and LUS diagnostic integration in detecting pulmonary contusion.

## 3. Confirming or Excluding Pneumothorax and Monitoring its Evolution

Pneumothorax is defined as the presence of air in the pleural space. It indicates the loss of visceral or parietal pleural membrane integrity, thereby allowing air from the environment or respiratory tract to accumulate in the pleural space. Although intrapleural pressures are negative throughout most of the respiratory cycle, air does not enter into the pleural space [27,56]. Iatrogenic pneumothorax is one of the main iatrogenic complications in ICU patients, and its occurrence increases the duration of ICU and hospital stays.

Iatrogenic pneumothorax occurs chiefly as a complication of barotrauma related to mechanical ventilation or as a post-procedural event (transthoracic needle biopsy, subclavian vein catheterization, thoracentesis, transbronchial, lung biopsy, pleural biopsy) [27,56]. The development of lung-protective strategies for ventilation and of new material, techniques, and recommendations for inserting central vein catheters (CVCs) makes iatrogenic pneumothorax largely preventable in routine practice. Nevertheless, its occurrence is closely related to the underlying disease, such as in patients with adult ARDS [27,56]. Other causes of pneumothorax such as spontaneous or traumatic-acquired may represent an additional portion of patients admitted to the ICU who need diagnosis and monitoring [27,56].

Symptoms of a pneumothorax can include chest pain, shortness of breath, cough, and increases in heart rate or breathing.

Different treatments are used depending on the size of the air volume in the pleural space and the amount of pressure it puts on the lung [27,56]. Small and asymptomatic iatrogenic pneumothoraxes often do not need any treatment and resolve spontaneously. Parenchymal lung damage can effectively heal, and the intrapleural air will be resorbed over time. In larger or symptomatic pneumothoraxes, simple manual aspiration or placement of a small catheter or chest tube attached to a Heimlich valve usually is successful. Tension pneumothorax presents as a life-threatening emergency that requires prompt recognition and treatment. When tension pneumothorax develops, urgent thoracic decompression is recommended [27,56]. For this reason, it is understandable how an early diagnosis can lead to the best therapeutic direction.

On the CXR, radiolucent air and the absence of juxtaposed lung markings between a compressed lobe or lung and the parietal pleura are indicative of pneumothorax. Tracheal deviation and mediastinal shift occur in a large tensive pneumothorax [41,42]. However, for the aforementioned diagnostic CXR limitations that are also due to the supine patient’s position, the risk of incurring an occult small pneumothorax is not negligible where the failure to recognize it is important, especially if the patient must undergo ventilation [27,56]. In such contexts, LUS offers an important contribution to the diagnosis and in the same way to monitoring.

On LUS, the most specific finding is the lung point, i.e., the boundary between the airy lung and pneumothorax visible as a change in physiological pleural sliding; the presence of air will also generate overlapping A-lines and the complete absence of B-lines [27,29,30].

Figure 14 shows an example of CXR and LUS diagnostic integration in detecting pneumothorax.

## 4. Checking and Monitoring the Devices

Medical devices, including ventilators, infusion pumps, breathing and feeding support, and others, improve patient care and outcomes in the ICU. Despite the staff’s ability to manage them, using them properly and checking their correct position reduces the development of adverse events associated with them [57]. The correct positioning of the various life-sustaining devices in the ICU represents a further diagnostic challenge in the correct reading of the CXR (Figure 15) [41].

In addition to the well-known operative use of LUS in positioning CVCs and tracheostomy procedures, LUS is an integrative diagnostic tool with CXR for monitoring thrombosis or pneumothorax complications related to CVCs and nasogastric tube displacements (Figure 16) [58,59,60].

## 5. Conclusions

LUS is a well-established diagnostic tool inextricably linked to the clinical environment and frequently used in ICUs. Despite their reduced diagnostic sensitivity in the anteroposterior supine projection, CXRs nevertheless remain an indispensable ‘open window’ to the chest that also provide information on parenchymal changes far from the pleura as well as on the cardio-mediastinum.

From a purely radiological point of view, in addition to a well-codified organization, the use of point-of-care LUS as an advanced method in ICU CXR reports can be an essential tool for diagnosing and monitoring areas of decreased transparency around the chest found on CXRs, such as atelectasis, pneumonia, pleural effusion, pulmonary edema, ARDS, and pulmonary contusions, as well as in diagnosing and monitoring pneumothorax and also when checking the positioning of the devices or the complications associated with them.

## Figures and Tables

**Figure 1 jpm-13-01541-f001:**
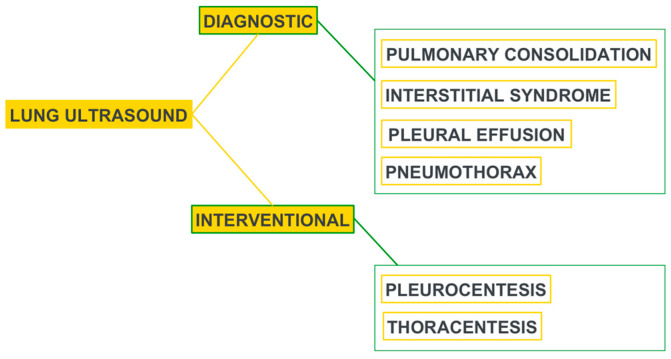
Utility of LUS in the ICU. Modified from [11].

**Figure 2 jpm-13-01541-f002:**
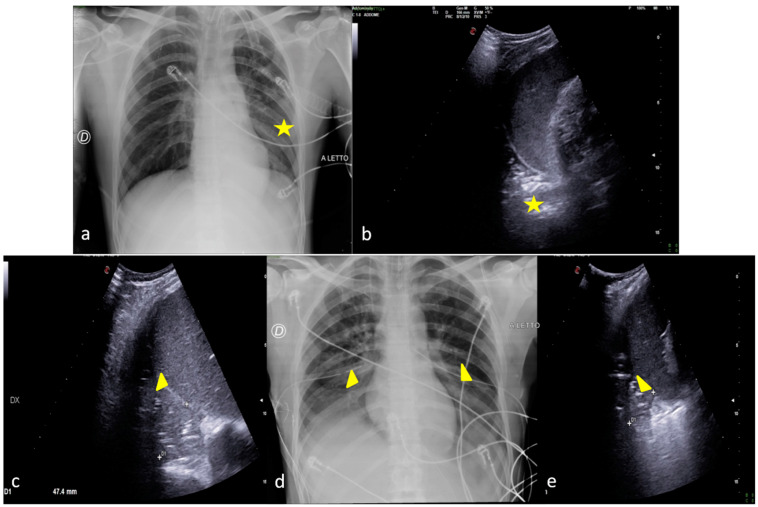
Two examples of CXR and LUS diagnostic integration. Bedside CXR (**a**,**d**) and LUS diagnostic integration (**b**,**c**,**e**). In the first case (top line) the CXR showed small, blurred opacities in the left inferior pulmonary field (**a**, star); LUS of the left basis confirmed the consolidative area with hyperechogenic spots as signs of an air bronchogram (**b**, star). In the second case (below line), the CXR showed small, blurred opacities in the inferior pulmonary field bilaterally (**d**, arrowhead); LUS confirmed areas of lung consolidation with an air bronchogram in the lower right and left pulmonary fields without pleural effusion (**c**,**e**, arrowhead).

**Figure 3 jpm-13-01541-f003:**
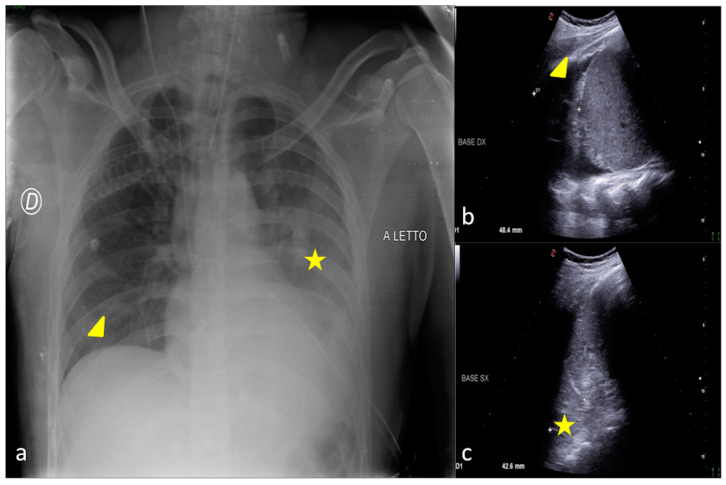
A 30-year-old male patient admitted to the ICU for motor vehicle crash polytrauma resulting in multiple costal fractures and coma status. Bedside CXR (**a**) and LUS (**b**,**c**). (**a**) The CXR showed a blurred opacity in the left inferior pulmonary field (star); the basal field of the right lung appeared normally expanded (arrowhead). (**c**,**d**) LUS diagnostic integration performed on the same day showed a fluid collection in the basal region of the right lung indicating the presence of a small pleural effusion (**b**, arrowhead) that was not clearly demonstrable in the bedside CXR and also an inhomogeneous hypo-echogenicity indicating a parenchymal consolidation in the basal region of the left lung without fluid collection (**c**, star).

**Figure 4 jpm-13-01541-f004:**
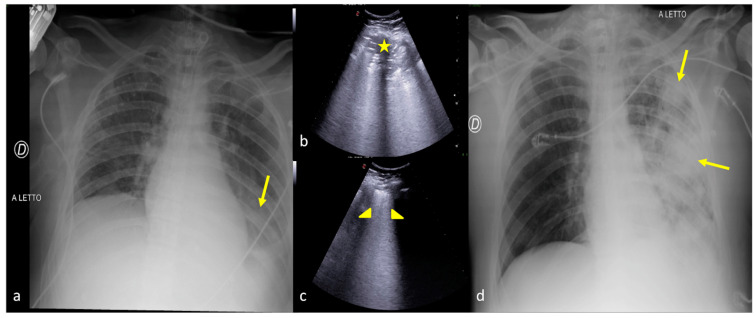
A 27-year-old male patient admitted to the ICU for high-grade gunshot trauma with abdominal involvement and a clinically worsened condition after intubation. Bedside CXR (**a**,**d**) and LUS (**b**,**c**). (**a**) The admission CXR showed a good expansion of the lungs with just a subtle and blurred opacity in the left inferior field (arrow). (**b**,**c**) LUS follow-up was performed after 2 days with evidence of hypoechogenic consolidative change in the left parenchyma (**b**, star) and a compact disposition of the B-lines as a sign of interstitial involvement (**c**, arrowhead) suggestive of phlogistic parenchymal complication. (**d**) The CXR confirmed the LUS findings showing some ovular opacity with a confluence trend occupying the left superior, middle, and inferior pulmonary fields (**d**, arrows).

**Figure 5 jpm-13-01541-f005:**
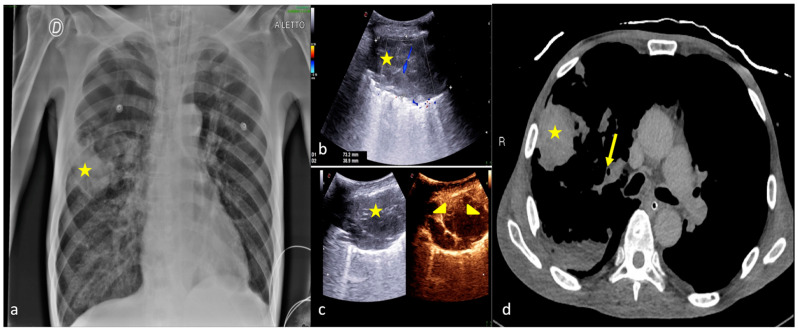
A 70-year-old male patient admitted to the ICU with acute respiratory failure from chronic obstructive pulmonary disease. Bedside CXR (**a**) LUS (**b**), CEUS (**c**), and CT scan (**d**). (**a**) The bedside CXR showed an area of pseudo-nodular consolidation (star) in the medium right pulmonary field. (**b**) LUS confirmed the parenchymal consolidation (star) that appears as a large heterogeneous hypoechoic area in the subpleural lung parenchyma. (**c**) CEUS examination with the administration of a sonographic contrast agent (Sonovue^®^, Bracco, Milan IT) showed no significative contrast enhancement in the pulmonary consolidation area (**c**, star) during the different phases of the study (**c**, arrowhead) concerning bronchial consolidation with the segmentary obstructive atelectatic area. (**d**) The CT scan highlighted areas of segmental parenchymal consolidation (**d**, star) caused by mucoid obstruction (**d**, arrow).

**Figure 6 jpm-13-01541-f006:**
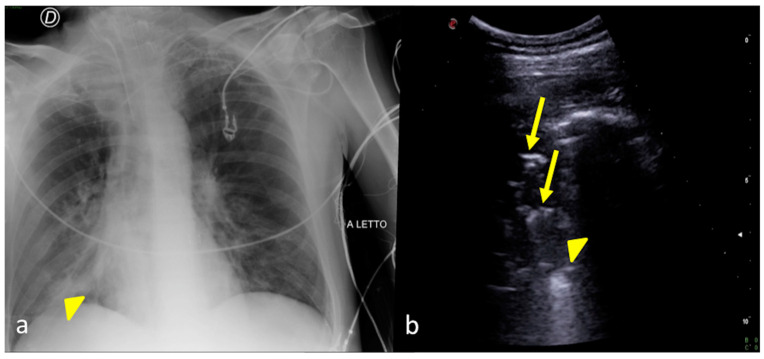
A 48-year-old male patient admitted to the ICU for post-traumatic subarachnoid hemorrhage and acute respiratory failure. Bedside CXR (**a**) and LUS (**b**). (**a**) The CXR showed an inhomogeneous opacity in the right inferior pulmonary field (arrowhead). (**b**) LUS confirmed a parenchymal consolidation area at the basis of the right lung (**b**, arrowhead) showing some hyperechogenic spots as signs of a static bronchogram (**b**, arrows) related to the diagnostic hypothesis of an atelectatic area.

**Figure 7 jpm-13-01541-f007:**
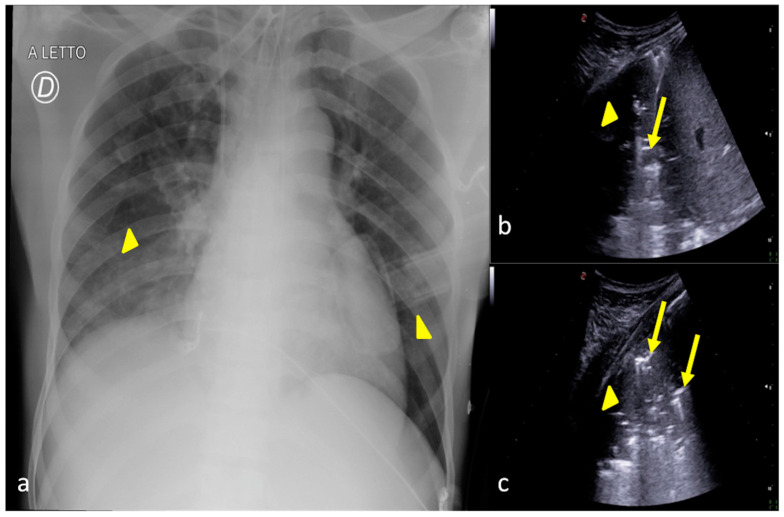
A 36-year-old female patient admitted to the ICU for a comatose state related to cerebral hemorrhage with fever and dyspnea after endotracheal intubation. Bedside CXR (**a**) and LUS (**b**,**c**). (**a**) The CXR showed faint areas of decreased parenchymal transparency in the right and left inferior pulmonary field (**a**, arrowheads); (**b**). LUS confirmed the consolidative areas at the lung basis (**b**,**c**, arrowheads) with hyperechogenic spots as signs of an air bronchogram (**b**,**c**, arrows) that moved in line with the respiratory excursion. The clinical scenario and imaging findings were suggestive of phlogistic bronchopneumonia.

**Figure 8 jpm-13-01541-f008:**
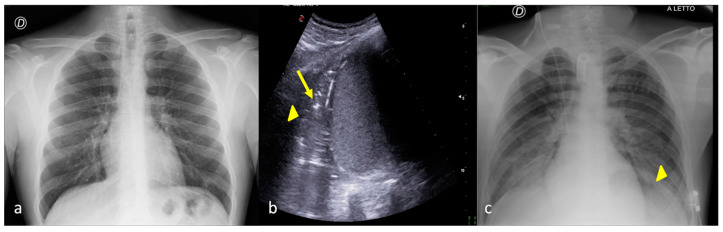
A 26-year-old male patient admitted to the ICU for a comatose state related to high-energy trauma due to a car accident. Bedside CXR (**a**,**c**) and LUS (**b**). (**a**) The CXR on the day of admission into the ICU showed normal lung expansion with no evidence of parenchymal change. (**b**) LSU was performed after 24 h endotracheal intubation with the onset of a respiratory worsening and showed an inhomogeneous area of mixed hypoechogenic change at the basis of the left lung (**b**, arrowhead) with some hyperechogenic spots suggestive of consolidation with an air bronchogram (**b**, arrow). (**c**) The CXR confirmed the LUS findings showing an area of reduced diaphony in the basal left field that was considered the manifestation of parenchymal consolidation (**c**, arrowhead). The clinical scenario and imaging findings were suggestive of phlogistic bronchopneumonia.

**Figure 9 jpm-13-01541-f009:**
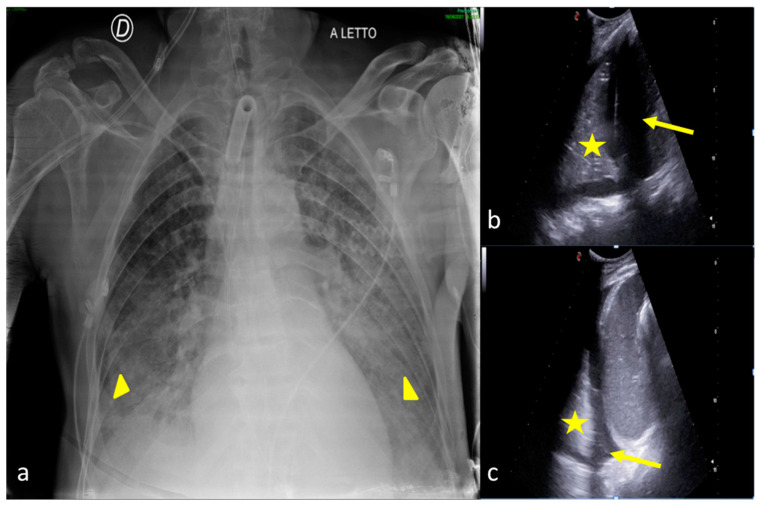
A 67-year-old male patient referred to the ICU after major trauma by a fall from a height. Bedside CXR (**a**) and LUS (**b**,**c**). (**a**) The CXR showed bilateral pulmonary opacities associated with pleural effusion (**a**, arrowheads). (**b**,**c**) Bilateral minimal pleural fluid was also confirmed by LUS (**b**,**c**, arrows) both on the right (**b**) and on the left (**c**) sides; lung consolidation was also visible (**b**,**c**, asterisks).

**Figure 10 jpm-13-01541-f010:**
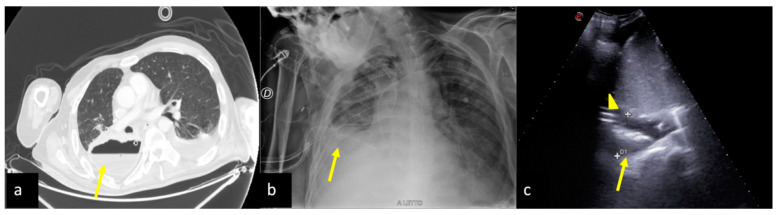
A 54-year-old male patient in respiratory failure. CT scan (**a**), CXR (**b**), and LUS (**c**). (**a**) The axial CT scan of the chest showed a parenchymal consolidation area with the air–fluid level in the basal segments of the right inferior pulmonary lobe suggestive of a pleural empyema (**a**, arrow). (**b**) The subsequent CXR control revealed a basal right-side decreased parenchymal transparency consistent with a persistent amount of the empyematous effusion (**b**, arrow). (**c**) LUS follow-up showed a better quantification of the residual fluid amount, with an inhomogeneous content of echogenic substance in suspension (**c**, arrow) and also confirmed the correct position of the surgical drainage tube (**c**, arrowhead).

**Figure 11 jpm-13-01541-f011:**
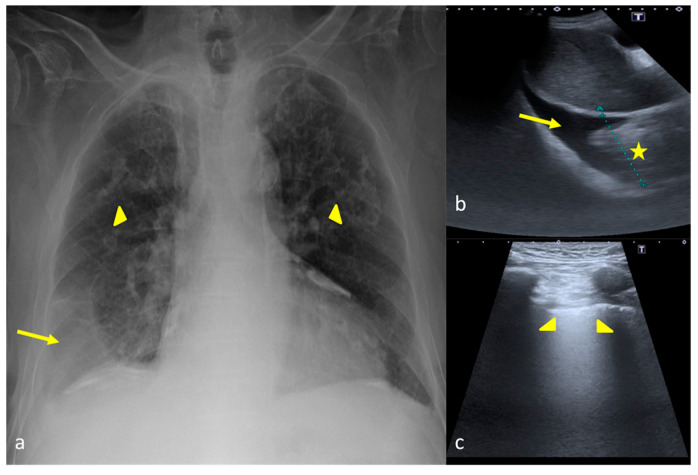
A 79-year-old male patient suspected of having pulmonary edema. CXR (**a**) and LUS (**b**,**c**). (**a**) On the CXR opacity of the middle and inferior field was detected bilaterally (**a**, arrowheads) with right lung basis consolidation (**a**, arrow). (**b**,**c**) LUS integration showed on the right lung basis (**b**) a pleural anechoic effusion (**b**, arrow) associated with a consolidative area (**b**, star); the evaluation of the right and left lungs (**c**) showed a compact appearance of the B-lines on the explorable lung areas in the intercostal space (**c**, arrowheads). In this case LUS allowed us to better clarify the nature of the CXR lung opacities.

**Figure 12 jpm-13-01541-f012:**
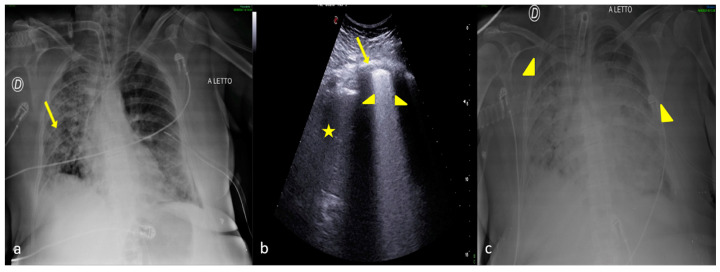
A 57-year-old female patient with acute respiratory failure during interstitial pneumonia not related to SARS-CoV2 infection. Bedside CXR (**a**,**c**) and LUS (**b**). (**a**) The CXR revealed on the right side multiple small opacities with signs of interstitial thickening suggestive of an interstitial and alveolar infective process (**a**, arrow). (**b**) The LUS examination performed after 5 days showed in the middle right pulmonary field and on the left side a compact appearance of B-lines (**b**, arrowheads) associated with some areas of parenchymal consolidation (**b**, star) with an irregular pleural line (**b**, arrow) that was suggestive of ARDS complication, according to the clinical worsening refractory to therapies. (**c**) The CXR confirmed massive involvement of the interstitial compartment and bilateral consolidation of the parenchyma with an internal air bronchogram suggestive of ARDS according to the clinical scenario (**c**, arrowheads).

**Figure 13 jpm-13-01541-f013:**
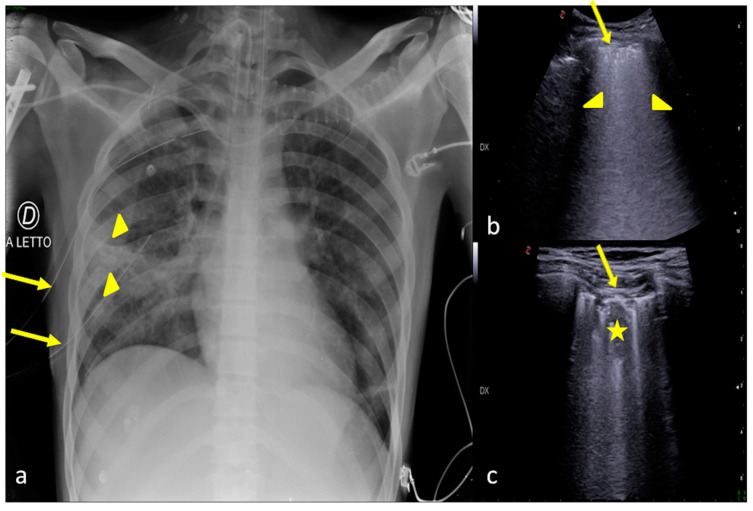
A 33-year-old male patient referred to the ICU after major trauma by defenestration. Bedside CXR (**a**) and LUS (**b**,**c**). (**a**) The CXR revealed on the right side the presence of two thoracic tube drainages (**a**, arrows) after pneumothorax (not shown) with middle and lower right lung zones indicating several confluent opacities (**a**, arrowheads). (**b**,**c**) LUS revealed on the same side the presence of a thick pleural line (**b**,**c**, arrow) with multiple B-line artifacts (**b**, arrowheads) and hypoechoic alveolar consolidation (**c**, star) corresponding to lung contusion.

**Figure 14 jpm-13-01541-f014:**
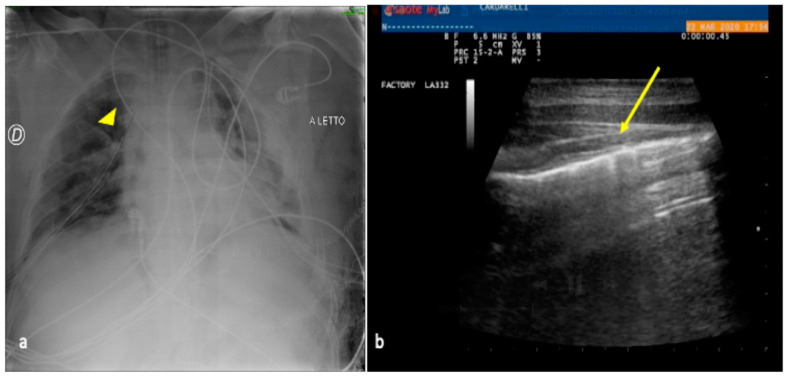
A 50-year-old male patient referred to the ICU after major trauma. Bedside CXR (**a**) and LUS (**b**). (**a**) The CXR showed the surgical drainage tube (arrowhead), with poor confidence on the entity of the residual pneumothorax. (**b**) LUS showed the presence of “lungs points” (arrow) confirming the residual pneumothorax that was not clearly demonstrable on the CXR.

**Figure 15 jpm-13-01541-f015:**
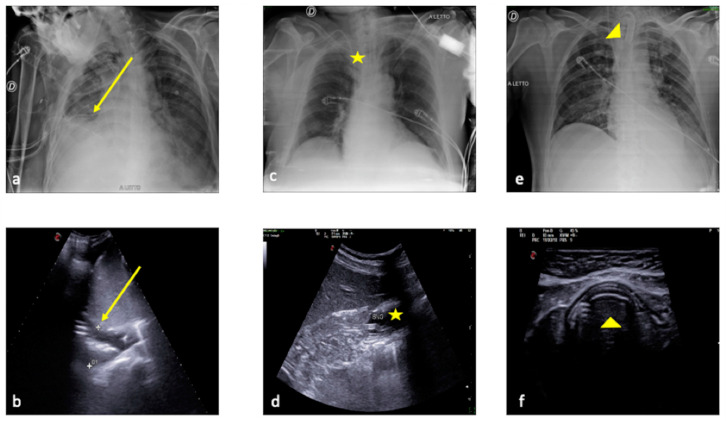
Comparison between plain chest radiography and LUS in three different critical patients in the intensive care unit. (**a**,**b**) CXR and LUS surgical drainage tube evaluation (the same patient as in Figure 10) (**a**,**b**, arrow). (**c**,**d**) CXR and LUS nasogastric tube evaluation (**c**,**d**, star). (**e**,**f**) CXR and LUS endotracheal tube evaluation (**e**,**f**, arrowhead).

**Figure 16 jpm-13-01541-f016:**
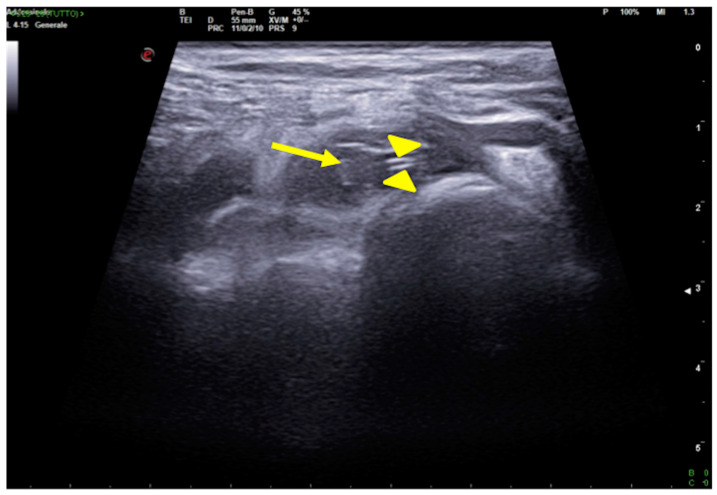
CVC thrombosis. Right supraclavicular ultrasound exploration highlights the presence of an isocogenic, non-compressible area attributable to thrombosis in the lumen of the subclavian vein (arrow); it also allows visualization of the course of the catheter, visible as a hyperechoic tubular structure (arrowheads).

**Table 1 jpm-13-01541-t001:** Validated clinical management LUS protocols.

Validated Clinical Management LUS Protocol	Purpose
BLUE protocol (Bedside Lung Ultrasonography in Emergency)	Emergency protocol for immediate diagnosis of acute respiratory failure.
FALLS protocol (Fluid Administration Limited by Lung Sonography)	Emergency protocol designed to sequentially rule out differential diagnoses such as cardiogenic and hypovolemic shock, allowing an early diagnosis of septic shock.
CAUSE protocol (Cardiac Arrest Ultrasound Exam)	Emergency protocol in cardiac arrest management. It has the potential to reduce the time required to determine the etiology of a cardiac arrest and thus decrease the time between arrest and appropriate therapy.

**Table 2 jpm-13-01541-t002:** Lung ultrasound and chest X-ray sensitivity and specificity in many lung diseases. Modified from [5].

Pulmonary Acute Disease	CXR	LUS
Pulmonary Consolidation	Sensitivity of plain chest radiography in detection of pulmonary consolidation has been reported as 38% to 76%.	Sensitivity and specificity of LUS for detection of pulmonary consolidation have been reported as 86–97% and 89–94%, respectively.
Interstitial Syndrome (Cardiogenic pulmonary edema, ARDS)	CXR showed a sensitivity of 36%, specificity of 90%, PPV of 29% and NPV of 92% while these results combined with clinical examination findings became 50%, 84%, 28% and 93% respectively.	US abnormalities may precede those of radiography and can be diagnostic, with a sensitivity and specificity of 97% and 95%, respectively.
Pleural Effusion	Supine chest radiography may reveal abnormality when the amount of fluid reaches 175–525 mL, which is higher than that for upright chest radiography.	US may detect 5–20 mL of pleural fluid with an overall sensitivity of 89–100% and specificity of 96–100%.
Pneumothorax	Portable chest radiography has a sensitivity of 19.8–31.8% and specificity of 99.3–100%.	The overall sensitivity and specificity of US in the detection of pneumothorax are 78.6–100% and 96.5–100%.

**Table 3 jpm-13-01541-t003:** ICU pleural–parenchymal and device complications grouped by CXR general findings with literature references reporting LUS-validated findings.

ICU Pulmonary Acute Disease	CXR General Findings	LUS-Validated Findings
Pulmonary Consolidation (Atelectasis/Pneumonia/Contusion)	Areas of decreased transparency	Yes [11,13,14,15,19,20,21,22]
Interstitial Syndrome (Cardiogenic pulmonary edema, ARDS)		Yes [13,14,15,22,23,24]
Pleural Effusion		Yes [11,13,14,20,25,26]
Pneumothorax	Areas of increased transparency	Yes [11,13,20,27]
**Device Complication**	Displacement	Empirical

## Data Availability

Not applicable.

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
