# Peer review of "Point-of-Care Lung Ultrasound in the Intensive Care Unit—The Dark Side of Radiology: Where Do We Stand?"

_jpm, 2023, doi:10.3390/jpm13111541_

Round 1
Reviewer 1 Report
Comments and Suggestions for Authors
This paper presents great review of topic of lung ultrasound in the hands of radiologist.
Especially in some countries, ultrasound is largely take over by other specialisations, out of the scope of radiology, and lung ultrasound is one of the points, where radiology is largely considered irrelevant in the topic. This paper proves otherwise.
I greatly appreciate the idea of combination (or maybe integration) of the chest X ray with lung ultrasound in one imaging approach, nicely demonstrated with pictorial essay. I can say that I will incorporate this approach in my own practice.
I have no serious or specific suggestions into the information or content of paper.
Comments on the Quality of English LanguageJust a few language issues:
- line 188: I would replace "drowned" with "shifted"
- line 328: word "hypodiaphania" - this word is actually used multiple times - I consider it highly specific regiolect, found specifically in works of Italian authors, it is not generally comprehensible, I would suggest "decreased transparency" instead
- line 367: "ipoechogenic" - possibly typo, from the context word "anechoic" should be used
- line 466: "air sac" - air volume or airspace would be more appropriate
- in some phrases, the the syntax of phrases is less authentic for english (possibly by influence of native language)
- line 485: "pleural flow" - I would suggest "pleural sliding"
Author Response
We thank the Reviewers for their valuable time dedicated to the improvement of this work and the precious suggestions they provided.
According to their suggestions, we have made the corrections to the highlighted sentences.
Reviewer 2 Report
Comments and Suggestions for Authors
This paper provides convincing examples of the utility of LUS in conjunction with CXR in the intensive care setting. I have a couple of minor comments:
1. Grammatical correction: in the abstract, "Although Computed Tomography (CT) can be con- 18 sidered the diagnostic gold standard in line with the clinical complexity of these patients who 19 usually require respiratory assistance and constant monitoring of vital functions and devices, however it is not a bedside diagnostic method.", should be "Although......devices, it is, however, not a ......"
2. The word "seems" is used twice in the Introduction section. I suggest using the word "is" instead, if the authors believe in their statements.
Comments on the Quality of English Language
Needs some editing to make the long sentences more readable.
Author Response

(The authors gave the same response as above.)
